# Overexpression of the Phosphoserine Phosphatase-Encoding Gene (*AtPSP1*) Promotes Starch Accumulation in *Lemna turionifera* 5511 under Sulfur Deficiency

**DOI:** 10.3390/plants12051012

**Published:** 2023-02-23

**Authors:** Lei Wang, Yingying Kuang, Siyu Zheng, Yana Tong, Yerong Zhu, Yong Wang

**Affiliations:** 1College of Life Science, Nankai University, Tianjin 300071, China; 2Tianjin Academy of Agricultural Sciences, Tianjin 300192, China

**Keywords:** starch accumulation, *AtPSP1* overexpression, sulfur deficiency, *Lemna turionifera* 5511

## Abstract

Duckweeds are well known for their high accumulation of starch under stress conditions, along with inhibited growth. The phosphorylation pathway of serine biosynthesis (PPSB) was reported as playing a vital role in linking the carbon, nitrogen, and sulfur metabolism in this plant. The overexpression of *AtPSP1*, the last key enzyme of the PPSB pathway in duckweed, was found to stimulate the accumulation of starch under sulfur-deficient conditions. The growth- and photosynthesis-related parameters were higher in the *AtPSP1* transgenic plants than in the WT. The transcriptional analysis showed that the expression of several genes in starch synthesis, TCA, and sulfur absorption, transportation, and assimilation was significantly up- or downregulated. The study suggests that *PSP* engineering could improve starch accumulation in *Lemna turionifera* 5511 by coordinating the carbon metabolism and sulfur assimilation under sulfur-deficient conditions.

## 1. Introduction

Duckweeds are the smallest flowering plant known to date and have severely reduced anatomies [1]. They are widely distributed all over the world, except in the Antarctic and Arctic regions [2]. There are 5 genera of duckweeds, namely *Spirodela*, *Landoltia*, *Lemna*, *Wolffiella*, and *Wolffia*, with a total of 36 species [3]. The biomass accumulation in duckweeds is much higher than the corn in dry weight (DW)/ha/year due to its asexual propagation and rapid growth [2]. The growth rate is attuned to the richness of the growth medium. It has been reported that duckweed can be converted into four different forms of energy, namely bio-oil, natural gas, bioethanol, and high-value-added industrial precursors, through different conversion technologies [4]. This makes duckweed a promising source of starch and a potential feedstock for the production of bioethanol and other biofuels [5]. The starch is mainly synthesized in the fronds of duckweeds, and 3–60% of dry weight can be accumulated when duckweeds are treated by growth regulators, heavy metals, nutrient deficiency, and salt stress [6,7,8,9,10,11,12,13,14,15,16,17,18,19]. However, these treatments always inhibit the growth of duckweed. The starch accumulation resulted from the decrease in duckweed biomass, while the sulfur deficiency was found to improve starch yields in duckweed without affecting its growth or biomass accumulation. Sulfur deprivation resulted in the highest starch yield, which was higher than the nitrogen or phosphorus limitation conditions. Previous research suggested that the cultivation of sulfur limitations is a potential strategy to prompt starch accumulation in duckweed [20].

In plants, three biosynthesis pathways of serine (Ser) have been described, which complicate the understanding of their metabolic regulation. One is the ethanoic acid pathway associated with photorespiration, and the other two alternative non-photorespiratory pathways of serine biosynthesis are the phosphorylation pathway and the glyceric acid pathway [21,22,23]. In most organisms, serine is synthesized via the phosphorylation pathway of serine biosynthesis (PPSB). Three enzymes are involved in the phosphorylation pathway of serine biosynthesis. First, 3PGA is oxidized to 3-phosphate hydroxypyruvate (3PHP) by 3-phosphoglycerate dehydrogenase (PGDH); then, 3PHP is converted to 3-phosphoserine via 3-phosphoserine aminotransferase (PSAT). Finally, 3-phosphoserine is dephosphorized to serine by 3-phosphoserine phosphatase (PSP) [24]. PSP was reported to be inhibited by high concentrations of serine; mutants lacking PSP in Arabidopsis are embryonically lethal, with altered pollen and chorion development, and *PSP* overexpression leads to increased nitrate reductase activity and photorespiration in leaves under light [25,26,27,28]. Metabolomic studies of *PSP1* overexpression and knockdown demonstrate that subtle changes in PPSB activity can modulate the glycolytic flux, affecting the TCA cycle and amino acid biosynthesis, which, in turn, affects glucose metabolism. These reports demonstrated that PSP plays a crucial role in plant metabolism and development, affecting glycolysis, amino acid synthesis, and the TCA cycle [28]. However, there are few studies on the relationship between PSP and starch biosynthesis in duckweed.

The sulfur element is an indispensable macronutrient to maintain normal growth in the plant. Sulfate is absorbed into root cells, transported to the plastids, and activated to form 5′-adenylylsulfate (APS) by ATP sulfurylase (ATPS); then, APS is reduced by APS reductase (APR) and sulfite reductase to sulfide; finally, sulfide and O-acetyl-L-serine (OAS) synthesize cysteine (Cys). Cys is the primary product of sulfur assimilation; its synthesis is dependent on the serine used to provide the carbon and nitrogen skeleton. Firstly, Ser is used with acetyl coenzyme A for the synthesis of O-acetyl serine (OAS) via serine acetyltransferase (SERAT). Then, O-acetyl serine (thiol) lyase (OAS-TL) replaces the activated acetyl portion of OAS with sulfide and releases cysteine [29,30]. Thus, Cys synthesis requires crosstalk between carbon and nitrogen metabolism and sulfur assimilation in the plant [31], while Ser is essential for the assimilation of sulfur.

PPSB synthesizes Ser in plastids from 3-phosphoglyceric acid (3-PGA). In duckweed, more 3-PGA was found to transfer to glucose and was used for starch biosynthesis after sulfur-deficiency treatment [20]. We are interested in whether there is a link between the phosphorylation pathway of serine (PPSB) and starch accumulation in duckweed. In this work, *AtPSP1,* encoding the key enzyme of PPSB in Arabidopsis, was overexpressed in duckweed when studying the effect on starch accumulation and growth under sulfur deficiency. The possible mechanism of starch accumulation was investigated in duckweed overexpressing *AtPSP1* under sulfur-deficient conditions.

## 2. Results

### 2.1. Generation of AtPSP1 Overexpressing Transgenic Lines

To explore the role that this pathway plays in starch accumulation and the interaction between PPSB and sulfur deficiency in duckweed, we constructed an *AtPSP1* overexpression vector based on pCAMBIA 1301, replacing the GUS coding sequence with the full-length CDS sequence (888 bp) of *AtPSP1* from Arabidopsis (Figure 1A). The vector was transformed into *Lemna turionifera* 5511 using *Agrobacterium tumefaciens EHA105*. The regenerated plants were screened with hygromycin to obtain resistant, transgenic plants. Nine transgenic lines with *AtPSP1* overexpression were confirmed via the PCR amplification of *AtPSP1* CDs (Figure 1B). Furthermore, all these lines were analyzed for semi-quantitative RT-PCR of the *AtPSP1* gene (Figure 1C), and three differentially expressed plants (Line 1/2/3, named PSP-1/PSP-2/PSP-3, respectively) were selected for quantitative RT-PCR assays; higher PSP enzyme activity was found in transgenic lines (Figure 1D,E).

### 2.2. Overexpression of AtPSP1 Increased Dry Weight under Sulfur Deficiency

Ten duckweed plants with similar growth conditions were selected from the wild-type or three transgenic lines under normal growth conditions, then cultured in a Datko (medium with the total nutrition) and sulfur-deficient medium. The phenotype of duckweed was observed and photos were taken on day 9. The results showed that there were no morphological differences between the transgenic lines and WT (Appendix A). The statistical results of relative growth rate, fresh weight, and dry weight at 3d, 6d, 9d, 12d, and 15d also indicate that the overexpression of *AtPSP1* did not show any significant differences compared with the WT in the Datko medium (Appendix A).

The fronds of duckweed only showed the etiolation phenotype after the duckweed was cultured under sulfur-deficient conditions, but there was no significant difference between the transgenic lines and WT (Figure 2A and Appendix A). Although the relative growth rate (RGR) and fresh weight of duckweed gradually increased with treatment, the transgenic lines and WT showed the same growth trend (Figure 2B,C). The dry weight of the transgenic lines was progressively higher than that of the WT after 3 days and significantly higher than the WT at 9d and 12d (Figure 2D). The overexpression of *AtPSP1* in duckweed increased the dry weight under sulfur-deficient conditions.

### 2.3. The Effect of the Overexpression of AtPSP1 on Photosynthetic Pigment and Chlorophyll Fluorescence Parameters under Sulfur Deficiency

The comparison of photosynthetic pigment content (chlorophyll a, chlorophyll b, total chlorophyll, and carotenoid) at 9d under Datko cultivation showed that there was no significant difference between the transgenic lines and WT (Appendix A). The chlorophyll fluorescence parameters, after 6 days of cultivation in Datko, showed that Fv/Fm (PSII maximum light quantum production), the relative electron transfer rate (rETR), and Y(II) (actual photosynthetic efficiency of PS II) also had no significant differences (Appendix A). Under sulfur-deficient conditions, the content of the photosynthetic pigment after 9 days of treatment and the chlorophyll fluorescence parameters after 6 days of treatment showed different trends in transgenic lines compared with WT (Figure 3). The photosynthetic pigment content (chlorophyll a, chlorophyll b, and total chlorophyll content) of the transgenic lines was significantly higher than that of the WT after 9 days of sulfur-deficient treatment, particularly PSP-2 (Figure 3A–C). The carotenoid content of the transgenic lines was also significantly higher than that of the WT, particularly *PSP-1* and *PSP-2* (Figure 3D). The value of Fv/Fm was significantly higher in the transgenic lines than in the WT. The value of Fv/Fm was the highest in *PSP-2* (Figure 3E). rETR and Y(II) of the transgenic lines were also significantly higher than those of the WT with the increase in light intensity (Figure 3F,G). The results indicated that the photosynthetic performance of the transgenic lines was significantly higher than that of the WT under sulfur-deficient conditions.

### 2.4. The Effect of the Overexpression of AtPSP1 on Starch, Sugar and Soluble Protein Contents under Sulfur Deficiency

The starch content was measured after treatment for 3, 6, 9, and 12 days in the transgenic lines and WT under Datko medium and sulfur-deficient conditions. The starch content of the duckweed gradually increased with treatment, from about 25 mg/g DW at 3d to 45 mg/g DW at 12d (Appendix A); the starch yield also gradually increased, from about 0.05 mg/flask at 3d to about 0.65 mg/flask at 12d in the Datko medium (Appendix A). Both the starch content and yield showed no significant difference between the transgenic lines and WT.

The starch content was compared after the samples were treated for 3, 6, 9, and 12 days under sulfur-deficient conditions. The results showed that the starch content increased from about 35 mg/g DW at 3d to about 200 mg/g DW at 12d, which was significantly higher in the transgenic lines than that of the WT at 6d and 9d in Figure 4A. The starch yield also increased from about 0.1 mg/flask at 3d to more than 2.0 mg/flask at 12d, and the starch yield in the transgenic lines was significantly higher than that of the WT at both 6d and 9d (Figure 4B).

The content of the total sugar and soluble protein was determined after 9 days under sulfur-deficient cultivations. The total sugar content was found to be 0.25 mg/g in the WT and 0.2 mg/g in the transgenic lines. The content of total sugar in the transgenic lines was significantly lower than that of the WT (Figure 4C). The soluble protein content was not significantly different between the transgenic lines and WT, with both ranging from 52 to 58 mg/g (Figure 4D). Starch accumulation was increased in the transgenic lines under sulfur-deficient conditions, while the total sugar content was reduced, possibly due to the conversion of sugars to starch.

### 2.5. The Effect of the Overexpression of AtPSP1 on the Expression of Sulfur Assimilation-Related Genes under Sulfur Deficiency

There are four SULTRs in Arabidopsis*: SULTR1* is responsible for the uptaking of sulfate from the rhizosphere, *LtSULTR2* and *LtSULTR3* play the role of a transition sulfate from root to shoot, and *LtSULTR4* is responsible for the transport of vascular sulfate [29,30,31]. Four genes were found to encode the sulfate transporter in duckweed: *LtSULTR1*, *LtSULTR2*, *LtSULTR3,* and *LtSULTR4*. The expression of *LtSULTR1*, *LtSULTR2*, *LtSULTR3*, and *LtSULTR4* in the transgenic lines was, on average, 14.5-, 3.2-, 5.65-, and 2.85-fold higher than that of the WT using qRT-PCR, respectively (Figure 5). The expression of *LtAPK1* (adenylyl sulfate kinase) was significantly upregulated 2.6–2.75-fold compared with the WT. The expression of *LtAPR1* (adenylyl sulfate reductase) was significantly higher than that of the WT, by about 3.35-fold; *LtSERAT1* (serine acetyltransferase coding gene) was also significantly upregulated in the transgenic lines, by 1.75–2.0-fold, compared with the WT. The expression of *LtSUOX1* (sulfite oxidase) was significantly upregulated by about 3.0-fold in all three transgenic lines compared with the WT. However, the expression of *LtSIR1* (sulfite reductase) was significantly lower than that of the WT, by about 0.3–0.4-fold; only the expression of *LtATPS1* (ATP sulfurylase) did not significantly differ between the transgenic lines and WT. The sulfur metabolic pathway was shown to be more active in the transgenic lines under sulfur-deficient conditions compared with normal growth conditions (Datko cultured condition).

### 2.6. The Effect of the Overexpression of AtPSP1 on the Expression of Starch Metabolism-Related Genes under Sulfur Deficiency

The overexpression of *AtPSP1* promoted starch accumulation in duckweed under sulfur-deficient conditions. To study the expression of the genes related to starch metabolism, the transcriptional level of the genes was analyzed using qRT-PCR after sulfur-deficiency treatment for 6 days. The results showed that the expression of the two genes encoding ADP-glucose pyrophosphorylase, *LtAPS1* and *LtAPL1*, was significantly higher than that of the WT, by about 1.65-fold and 1.4-fold, respectively. The expression of *LtSSS1* (soluble starch synthase) was significantly higher in the three transgenic lines than in the WT, by about 4.5-, 5.2-, and 4.6-fold. The expression of *LtGBSS1*, the gene-encoded granule-bound starch synthase, was not significantly different compared with the WT, while the genes involved in the regulation of amylolytic metabolism showed different expression patterns. The expression of *LtISA1* (isoamylase) was significantly lower than that of the WT, by about 0.15-fold; the expression of *Ltα-Amy1*(α-amylase) was not significantly different compared with the WT; and *Ltβ-Amy1*(β-amylase) showed different expression patterns in the three transgenic lines—they were 0.7-fold, 0.85-fold, and 0.8-fold higher in the *PSP-1, PSP-2*, and *PSP-3* lines, respectively, than in the WT (Figure 6). The results showed that the expression of the genes involved in the starch synthesis pathway was upregulated, and the expression of the genes involved in the starch degradation pathway was downregulated.

### 2.7. The Effect of the Overexpression of AtPSP1 on the Expression of Carbon Metabolism-Related Genes under Sulfur Deficiency

The synthesis and degradation of plant starch are closely linked to carbon metabolism [32,33,34]; we examined the expression of the genes related to carbon metabolism in duckweed after 6 days of treatment under sulfur deficiency. According to the qRT-PCR results shown in Figure 7, the expression of *LtPFK1* (phosphofructokinase) in the glycolytic pathway was not significantly different between the transgenic lines and WT. In contrast, the expression of *LtFBA1* (fructose-diphosphate aldolase), *LtPK1* (pyruvate kinase encoding gene), and *LtPDC1* (pyruvate dehydrogenase complex) was significantly upregulated, by about 3.3-fold, 2.6-fold, and 2.4-fold, respectively, compared with WT. The expression patterns of the genes in the TCA cycle also differed. The expression of *LtACO1* (aconitase) was significantly upregulated in the transgenic lines, by about 2.4 times. The expression of *LtIDH1* (isocitrate dehydrogenase) was significantly higher than the WT, by about 1.2-fold. The expression of *Lt2OG-DH1* (2-OG dehydrogenase) was significantly higher in the three transgenic lines than in the WT, by about 2.6-fold, 2.25-fold, and 2.3-fold, respectively. The expression of *LtSDH1* (succinate dehydrogenase) in the three transgenic lines was, on average, 4.1-fold higher than in the WT. The expression of *LtFUM1* (fumarate) was also significantly higher than that of the WT, by about 3.3-fold, 2.75-fold, and 3.0-fold, in *PSP-1*, *PSP-2*, and *PSP-3*, respectively. The expression of *LtMDH1* (malate dehydrogenase) and *LtCS1* (citrate synthase) was upregulated compared with the WT but did not reach a significant difference. This indicated that the overexpression of *AtPSP1* increased the expression of carbon metabolism-related genes under sulfur-deficient conditions.

## 3. Discussion

### 3.1. The Overexpression of AtPSP1 Prompts Growth and Starch Accumulation in Duckweed under Sulfur Deficiency

Duckweed is rich in starch under normal growth conditions and can accumulate more starch when its growth is inhibited due to nutrient limitation, hormone treatments, etc. [6,7,8,9,10,11,12,13,14,15,35]. In recent years, much progress has been made regarding the mechanisms of starch accumulation in duckweed under nutrition stress conditions. It is believed that there is a tradeoff between starch accumulation and an increase in biomass or dry weight. In this work, the sulfur-deficient incubation did not cause morphological changes in the WT and *AtPSP1* expression lines. There was a reduction in the relative growth rate (RGR), fresh weight, and dry weight, as well as the content of chlorophyll and chlorophyll fluorescence parameters, in both the WT and transgenic duckweed, compared with normal growth conditions (Datko cultivation) (Figure 2 and Appendix A). The above results indicated that sulfur deficiency affected the growth of the duckweed. The results are in accordance with former reports on the morphological observations and growth parameters after duckweed is cultivated under nutrient limitations [20,35,36,37].

Interestingly, although there were no significant differences in RGR and fresh weight between the transgenic lines and WT, the dry weight was significantly higher than that of the WT after 6 days under sulfur-deficient cultivation. The photosynthetic pigment content was also significantly higher than that of the WT, as well as the chlorophyll fluorescence parameters, indicating that the photosynthetic function was better in the transgenic lines than in the WT. This explains the accumulation of more dry matter, which is similar to the report on *Spirodela polyrhiza* under sulfur-deficient conditions [35]. Sulfur-deficient cultivation promoted starch accumulation in duckweed when compared to Datko cultivation (Figure 4A,B and Appendix A), and the starch content was significantly higher in the transgenic lines than in the WT, indicating that the overexpression of *AtPSP1* could promote starch accumulation under sulfur-deficient conditions. The sugar content was lower than that of the WT in the transgenic duckweed, and there was almost no difference in protein content in both the transgenic duckweed and WT under sulfur limitation (Figure 4). There was a lower content of lipids and cellulose in duckweed. These results demonstrated that the dry weight increase was mainly caused by starch accumulation, which also suggested that there was no tradeoff between the accumulation of starch and dry weight under sulfur deficiency [20].

### 3.2. The Overexpression of AtPSP1 Coordinates Sulfur Assimilation, Starch Synthesis, and Carbon Metabolism under Sulfur Deficiency

How does the *AtPSP1* transgenic plant prompt starch accumulation under sulfur-deficient conditions? The transcriptional analysis of the genes related to sulfur assimilation showed that the overexpression of *AtPSP1* increased the expression of the genes involved in the process of sulfur assimilation, including the absorption, activation, and reduction of sulfur (Figure 5). The expression of four genes encoding the sulfate transporter, *LtSULTR1*, *LtSULTR2*, *LtSULTR3,* and *LtSULTR4,* was significantly upregulated. It has been reported that the plant could activate sulfate absorption from the rhizosphere by inducing the gene expression of high-affinity sulfate transporters (SULTR1) under sulfur limitations. Meanwhile, the plant could retrieve the sulfate stored in the vacuole by inducing the expression of SULTR4 and activate the translocation of sulfate from root to shoot by inducing the expression of SULTR2 and SULTR3 under sulfur limitation [31,35]. Our results indicated that the transgenic lines accelerated sulfate mobilization in duckweed to maintain the sulfur required for its growth. In addition to this, the increased expression of *LtSUOX1* could probably enhance the oxidation of endogenous sulfite to sulfate in the transgenic lines; the upregulation of the *LtAPR1* expression may increase the run-up of adenosine sulfate (APS) to sulfite; and the upregulation of *LtAPK1* expression presumably prompted the run-up of APS to PAPS (adenosine-5′-phosphoryl sulfate 3′-phosphate). With serine serving as a precursor of Cys biosynthesis, the increased expression of *LtSEART1* accelerated the conversion of serine to Cys (Figure 8). The expression mode of *LtSULTR1* and *LtSULTR4* is similar to the work on *Spirodela polyrhiza* under sulfur deficiency, as they were all upregulated [20], while the expression of the other genes involved in sulfur assimilation was between *Spirodela polyrhiza* and transgenic *AtPSP1* lines. Methionine (Met) and cysteine (Cys), which all contain sulfur, serve as essential compounds for plant growth and reproduction [38]. Serine is a precursor substance for their synthesis, and the mutation of *AtPSP1* was found to alter the sulfur metabolism in Arabidopsis [39]. This suggested that the overexpression of *AtPSP1* may play an important role in the prompting of sulfur assimilation, which could accelerate endogenous sulfur utilization to maintain duckweed growth.

The synthesis of starch is dependent on the genes involved in the starch synthesis pathway and degradation pathway. The expression of genes related to the starch synthesis and degradation pathway was examined under sulfur-deficient conditions (Figure 6). It was found that the expression of *LtAPS1*, *LtSSS1*, and *LtAPL1* was significantly higher in the starch synthesis pathway in the transgenic lines than in the WT, and the expression of *LtISA1* and *Ltβ-Amy1,* involved in the starch degradation pathway, was significantly lower than that of the WT. The expression pattern of the above genes is similar to the patterns found in *Lemna turionifera* 5511 under nitrogen starvation, which suggested that the accumulation of starch was probably a result of the upregulation of starch-synthesis-related genes and the downregulation of starch-degradation-related genes [15]. The synthesis of starch in plants is associated with the breakdown of substances such as sugars. The total sugar content in the transgenic lines was significantly lower than that of the WT under sulfur-deficient conditions (Figure 4C). This demonstrated that the increase in starch content was partly due to the conversion of sugars to starch, as previously found in *Lemna turionifera* 5511 under nitrogen starvation [15].

Plant starch biosynthesis is closely linked to carbon metabolic pathways, and the previous findings suggest that the process of starch accumulation affects the glycolytic and TCA cycle pathways [13]. The expression of the analyzed genes was involved in the glycolytic and TCA cycle pathways under sulfur deficiency (Figure 7). The glucose 6-phosphate is the main substance linking starch synthesis and the glycolytic pathway because the synthesis of starch requires the consumption of more glucose 6-phosphate. The expression of glycolysis-related genes, *LtFBA1*, *LtPK1*, and *LtPDC1,* was found to be significantly upregulated, which would increase the activity of the glycolytic pathway in the transgenic lines to maintain the balance of carbon metabolism in plants, including the accumulation of starch and the increase in the dry matter. The TCA cycle is one of the core pathways in plants; this is a downstream pathway linked to glycolysis and is closely related to the biosynthesis of many amino acids. In this work, the transcriptional level of TCA-cycle-related genes (mainly *LtACO1*, *LtIDH1*, *LtOG-DH1*, and *LtSDH1*) were significantly increased in *AtPSP1*-transformed duckweed under sulfur-deficient conditions, as observed in Arabidopsis when cultivated under a variety of stresses [40,41]. The results demonstrated that the *AtPSP1* overexpression prompted the link between the TCA cycle, carbon metabolism, and sulfur assimilation under sulfur-deficient conditions. 

## 4. Materials and Methods

### 4.1. Duckweed Culture and Sulfur-Deficiency Treatment

*Lemna turionifera* 5511 was grown in a sterile Datko medium, according to Yang [42]. Duckweed was cultured under long-day conditions (16 h light/8 h dark cycle) at 22 ± 2 °C and 100 μmol photons m^−2^·s^−1^ light intensity. Initially, fresh duckweed was grown in the Datko medium for 7 days. Then, 9 plants were transferred to the sulfur-deficient Datko medium and fresh Datko medium in 100 mL conical flasks. Plants were cultivated under the same long-day conditions and temperature, and the horizontal light intensity of the plants was 100 μmol photons m^−2^·s^−1^.

### 4.2. Vector Construction and Acquisition of Transgenic Duckweed

The known sequence of *AtPSP1* (the only gene encoding 3-phosphoserine phosphatase) was obtained from Arabidopsis cDNA using PCR. The pCAMBIA1301 plasmid containing the hygromycin resistance gene was inserted, the 5′ end was modified with the CaMV-35S promoter, and the 3′ end was the nopaline synthase gene terminator. The plant binary expression vector p*CAMBIA1301:35S:AtPSP1* was constructed. The vector was transformed into the Agrobacterium tumefaciens strain *EHA105* using the freeze–thaw method. The *AtPSP1* transgenic duckweed was obtained using the method of the callus transformation system described by Yang et al. [43]. DNA was isolated using the SDS method described by Yang et al. [43]. RNA was isolated using an Eastep Super Total RNA Extraction Kit (Promega, Shanghai, China) and reverse transcription to cDNA using a PrimeScriptTMRT reagent Kit with gDNA Eraser (TaKaRa, Beijing, China). The DNA and cDNA for *Lt18s* and *AtPSP1* were amplified using PCR and RT-PCR with specific primers (provided in Appendix A) to identify the transgenic duckweed.

### 4.3. Measurement of the PSP Enzyme Activity

PSP activity was determined by measuring the phosphate release of 3-phosphoserine at 25 °C in 0.2 mL reaction mixture containing 80 mM Tris-HCl (pH 7.5), 10 mM 3-phosphoserine, 3 mM MgCl_2_, and 160 μL of extract start the reaction. After 30 min, 0.25 mL of 15% trichloroacetic acid was added to stop the reaction. The concentration of free phosphate was determined, and one enzyme unit catalyzed the conversion of 1 mmol of substrate/min under specified conditions [27].

### 4.4. Biomass and Relative Growth Rate Determination

Ten duckweed lines were cultured in a 100 mL conical flask with the culture solution, and those treated for different growing days were pumped to dryness and weighed at fresh weight immediately, and then dried to constant weight in a hot-air-circulating oven at 80 °C and weighed as dry weight of the flask-cultured materials. 

Duckweed’s relative growth rate was calculated from the changes in fresh weight, and the 10 duckweed lines were weighed at the beginning growth day (*d_*0*_* = 0 d) and at the end (*d_x_* = 3, 6, 9, and 12 d) of the experimental period. The relative growth rate (RGR) was calculated according to Su [44].
RGR = (ln*Nj* − ln*Ni*)/Δ*t*(1)
ln represents the natural logarithm, *Nj* and *Ni* represent the fresh weight at time *j* and time *i*, respectively, and Δ*t* is the growth day instead of *d_x_ − d*_0_.

### 4.5. Determination of Photosynthetic Pigments and Chlorophyll Fluorescence Parameters

A total of 0.1 g of duckweed material was treated for 9 days in 1 mL of 95% ethanol and placed in an incubator at 28 °C until the duckweed turned completely white. After taking 200 μL of extract, the OD values at 663 nm, 645 nm, and 440 nm were measured under a plate reader, and the content of photosynthetic pigments was calculated according to the Arnon method [45]. Chlorophyll fluorescence parameters were determined using Mini-PAM-II (WALZ, Effeltrith, Germany). The values of Fv/Fm, rETR, and Y(II) of photosystem II were measured using the WALZ protocol.

### 4.6. Starch, Total Sugar, and Soluble Protein Determination

A starch extraction and measurement kit was used to extract and measure starch content according to the manufacturer’s protocol (BC0700, Solarbio Biological and Technology Co., Ltd., Beijing, China). The total sugar was extracted and measured using a total sugar extraction and measurement kit according to the manufacturer’s protocol (BC2715, Solarbio Biological and Technology Co., Ltd., Beijing, China). The soluble protein content extraction was measured by using 30 mM Tris-HCl (pH 7.0) according to the manufacturer’s protocol (P0006C, Beyotime Biotechnology Co., Ltd., Shanghai, China).

### 4.7. DNA and RNA Extraction and qRT-PCR

RNA was isolated using an Eastep Super Total RNA Extraction Kit (Promega, Shanghai, China). After reverse transcription, cDNAs for *Lt18s*, *AtPSP1*, starch metabolism genes, sulfur metabolism genes, and carbon metabolism genes were amplified using qRT-PCR and the specific primers designed from the duckweed genome sequence obtained through RNA-Seq (in Appendix A). qRT-PCR was performed on an iCycler thermal cycler (Bio-Rad iQ5, Hercules, CA, USA) using TB Green Premix Ex TaqII (code RR420A: TB Green Premix Ex Taq TM, Dalian Bao, Dalian, China) according to the manufacturer’s protocol. The reaction mixture was heated to 95 °C for 30 s, and then 40 PCR cycles were carried out, at 95 °C for 5 s, 58 °C for 30 s, and 72 °C for 30 s. After normalizing the data to *Lt18s*, differences in the relative expression levels of the detected genes were calculated using the 2^−ΔΔCT^ method. All values are shown as the mean ± standard error of the mean using at least three biological replicates.

### 4.8. Data Analysis

All data were measured in at least three biological replicates, and the experiments were repeated at least three times independently. Data were organized using Excel, statistically analyzed using SPSS 22.0 (IBM, Chicago, IL, USA), analyzed for significance using one-way ANOVA (* *p* < 0.05 and ** *p* < 0.01), and graphed using GraphPad 9.2 software (GraphPad Software, San Diego, CA, USA).

## 5. Conclusions

Duckweed is a promising feedstock for bioenergy. We found that the overexpression of *AtPSP1,* the last key enzyme of PPSB, can increase the dry weight and starch accumulation in duckweed, accompanied by a higher photosynthetic capacity under sulfur-deficient conditions. The transcriptional analysis demonstrated that better growth and starch accumulation resulted from coordinating the expression of the key enzymes involved in sulfur assimilation, starch synthesis, and metabolism, as well as the glycolytic pathways and TCA cycle in *AtPSP1*-overexpressing lines under sulfur deficiency. This study provides a prospective pathway for starch accumulation through genetic engineering and sheds light on the potential mechanism of PPSB-promoting starch accumulation under sulfur-deficient conditions.

## Figures and Tables

**Figure 1 plants-12-01012-f001:**
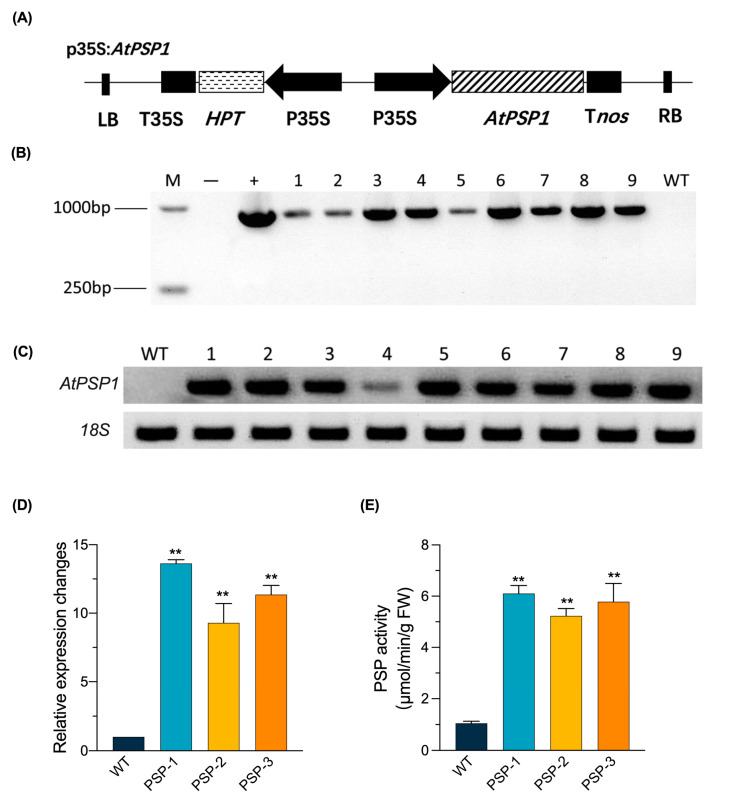
Construction of pCAMBIA1301-*AtPSP1* vector and identified the *AtPSP1* overexpressors lines: (**A**) T-DNA constructs designed for the overexpression of *AtPSP1*. LB, left border; *HPT*, encoding hygromycin-resistant gene; T35S and P35S, terminator and promoter of CaMV35S, respectively; T*nos*, terminator of *NOS*; RB, right border; (**B**) DNA-PCR identification of *AtPSP1* transformation, *AtPSP1* CDs was 888bp; (**C**) semi-quantitative RT-PCR analysis of *AtPSP1* transcripts with *18S* as reference gene; *18S* is *18S* ribosomal RNA of *Lemna turionifera* 5511; (**D**) real-time quantitative PCR analysis of *AtPSP1* transcript with *18S* as reference gene; (**E**) analysis of PSP enzyme activity. “M”, DL15000 DNA ladder; “−”, water as the PCR temple of negative control; “+”, plastid as PCR temple of positive control; WT, wild-type; PSP-1, PSP-2, and PSP-3, the three different *AtPSP1* overexpressing lines. Values given in (**D**,**E**) are the means ± standard error of 3 independent experiments with 3 repeats. The double asterisks (**) indicate significant differences (*p* < 0.01) from controls based on one-way ANOVA.

**Figure 2 plants-12-01012-f002:**
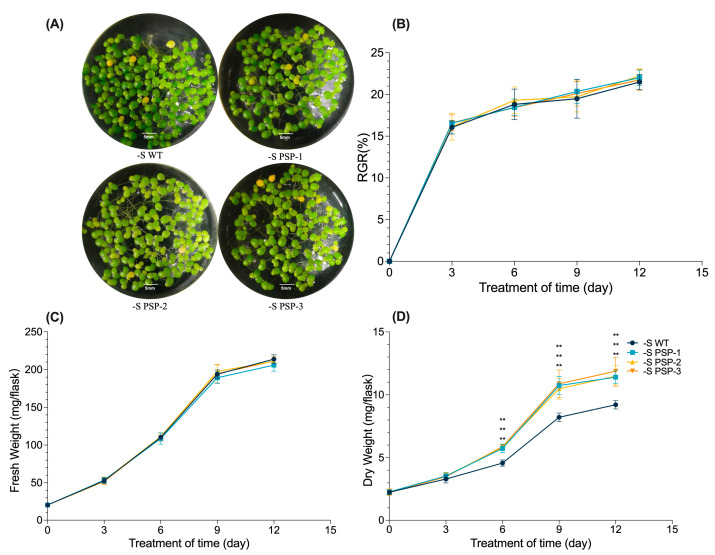
Analysis of phenotype, relative growth rate, fresh weight, and dry weight under sulfur-deficient conditions: (**A**) phenotypes of WT and three *AtPSP1* overexpressors; photos were taken after WT and *AtPSP1* transgenic plants were cultivated under sulfur-deficient conditions for 9 days; (**B**) analysis of relative growth rate (RGR); (**C**) comparison of fresh weight between WT and three *AtPSP1* overexpressors; (**D**) comparison of dry weight between WT and three *AtPSP1* overexpressors. The samples were harvested to analyze the growth rate, fresh weight, and dry weight after WT and *AtPSP1* transgenic plants were cultivated under sulfur-deficient conditions for 3, 6, 9, and 12 days, respectively. Each statistic is the mean ± standard error (*n* = 9, from 3 independent experiments with 3 repeats of each). The double asterisk symbol (**) indicates that the difference (*p* < 0.01), compared with the control, was significant, based on one-way ANOVA.

**Figure 3 plants-12-01012-f003:**
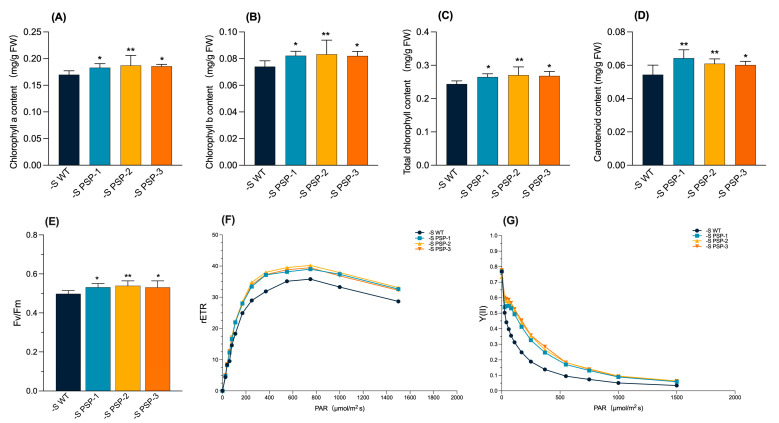
Analysis of photosynthetic pigment and chlorophyll fluorescence parameters in *AtPSP1*-transformed duckweed lines under sulfur-deficient cultivation: (**A**) chlorophyll a content; (**B**) chlorophyll b content; (**C**) total chlorophyll content; (**D**) carotenoid content. The photosynthetic pigment contents were measured after the samples were cultivated under sulfur deficiency for 9 days. (**E**), Fv/Fm; (**F**) rETR; (**G**) Y(II). Chlorophyll fluorescence parameters were measured after the samples were cultivated under sulfur deficiency for 6 days. Each statistic is the mean ± standard error (*n* = 9, from 3 independent experiments with 3 repeats of each, no average values in (**F**,**G**)). The asterisk symbol (*) represents statistically significant differences (*p* < 0.05); the double asterisk symbol (**) indicates that the difference (*p* < 0.01), compared with the control, was significant, based on one-way ANOVA.

**Figure 4 plants-12-01012-f004:**
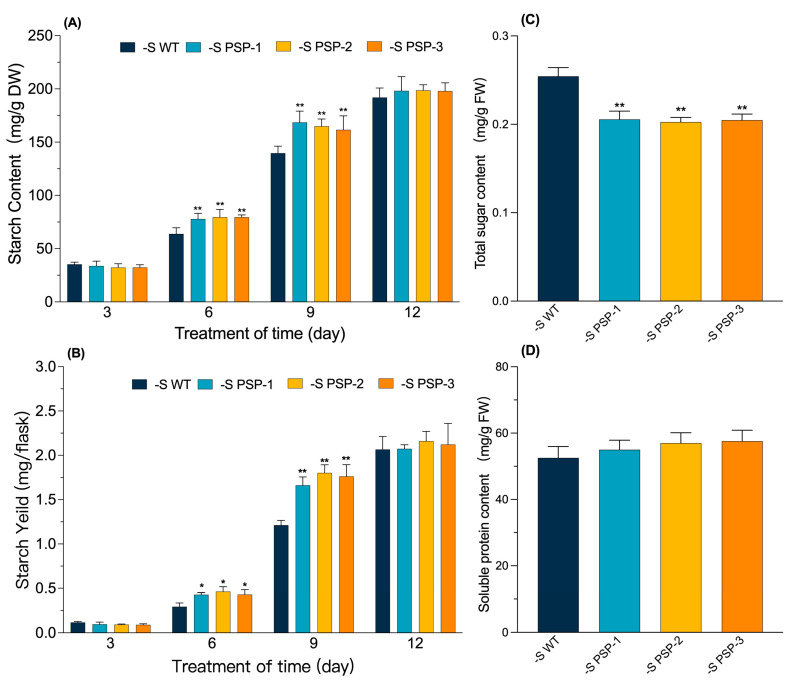
Effect on starch, total sugars, and soluble protein content of *AtPSP1*-transformed duckweed lines under sulfur deficiency, cultured for 9 days: (**A**) the starch content per gram dry weight; (**B**) the starch yield; (**C**) total sugar content; (**D**) soluble protein content. Each statistic is the mean ± standard error (*n* = 9, from 3 independent experiments with 3 repeats of each). The asterisk symbol (*) represents statistically significant differences (*p* < 0.05); the double asterisk symbol (**) indicates that the difference (*p* < 0.01), compared with the control, was significant based on one-way ANOVA.

**Figure 5 plants-12-01012-f005:**
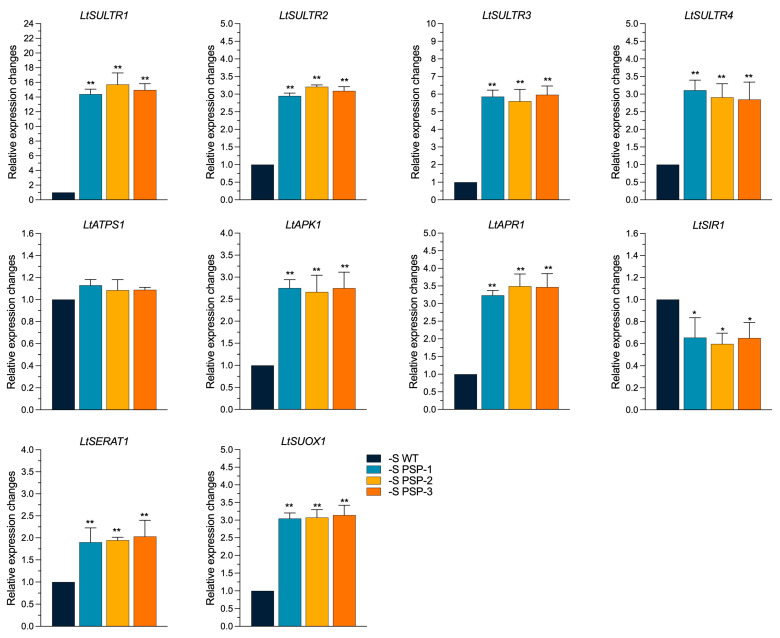
Effect on the expression of sulfur-assimilation-related genes in *AtPSP1*-transformed duckweed lines under sulfur-deficient conditions for 6 days. The relative expression level of the WT was normalized to 1. Each statistic is the mean ± standard error (*n* = 9, from 3 independent experiments with 3 repeats of each). The asterisk symbol (*) represents statistically significant differences (*p* < 0.05); the double asterisk symbol (**) indicates that the difference (*p* < 0.01), compared with the control, was significant, based on one-way ANOVA.

**Figure 6 plants-12-01012-f006:**
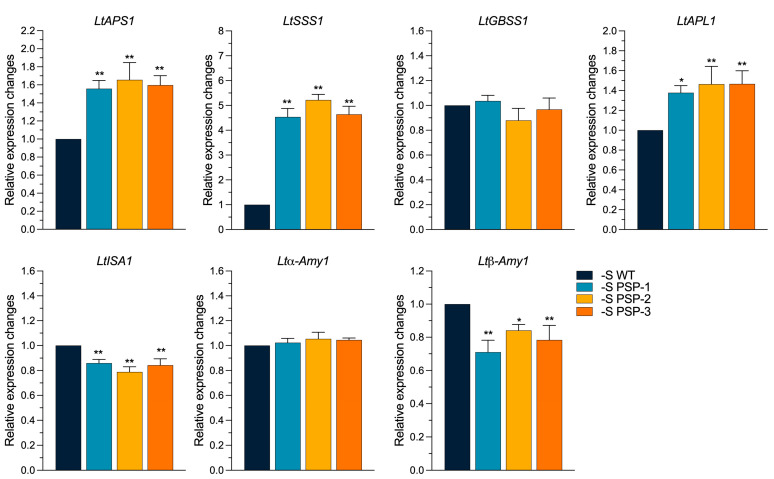
Effect on the expression of starch metabolism-related genes in *AtPSP1*-transformed duckweed lines under sulfur deficiency, cultured for 6 days. The relative expression level of the WT was normalized to 1. Each statistic is the mean ± standard error (*n* = 9, from three independent experiments with three repeats of each). The asterisk symbol (*) represents statistically significant differences (*p* < 0.05); the double asterisk symbol (**) indicates that the difference (*p* < 0.01), compared with the control, was significant based on one-way ANOVA.

**Figure 7 plants-12-01012-f007:**
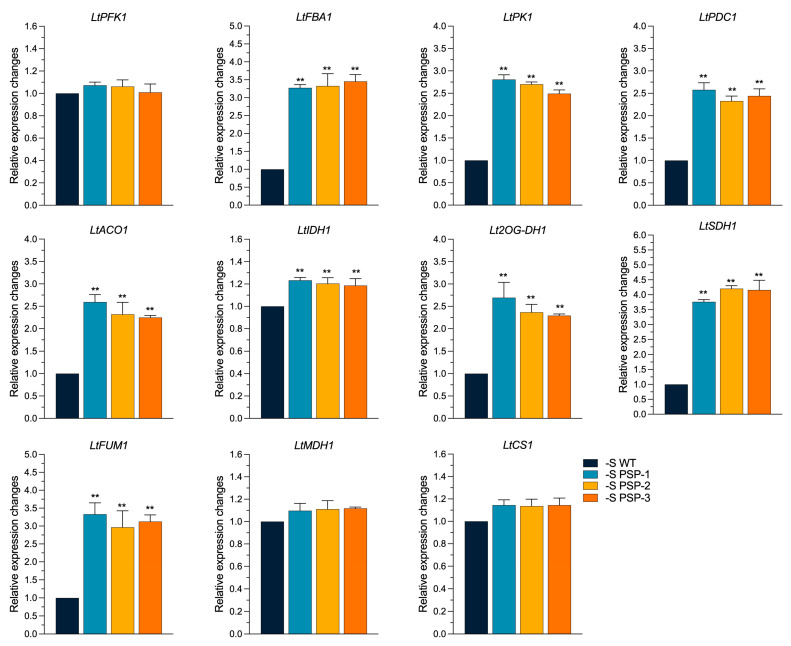
Effect on the expression of carbon metabolism-related genes in *AtPSP1*-transformed duckweed lines under sulfur deficiency, cultured for 6 days. The relative expression level of the WT was normalized to 1. Each statistic is the mean ± standard error (*n* = 9, from 3 independent experiments with 3 repeats of each). The asterisk symbol (*) represents statistically significant differences (*p* < 0.05); the double asterisk symbol (**) indicates that the difference (*p* < 0.01), compared with the control, was significant based on one-way ANOVA.

**Figure 8 plants-12-01012-f008:**
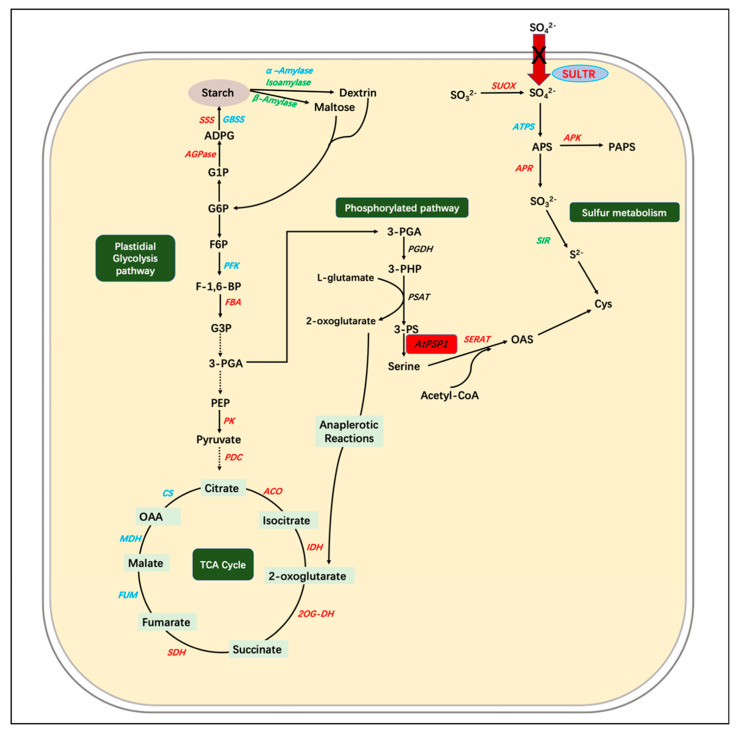
Expression pattern of genes involved in serine phosphorylated synthesis pathway, carbon metabolism, starch metabolism, and sulfur assimilation under sulfur deficiency in transgenic duckweed. The arrows indicate the directions of catalytic reactions or transportations. Red indicates upregulated expression, green indicates downregulated expression, and blue indicates no significant change. Dotted black arrows indicate omitted steps. AGPase, ADP−glucose pyrophosphorylase; ADPG, adenosine-5−diphosphoglucose; GBSS, granule−bound starch synthase; SSS, soluble starch synthase; G1P, glucose 1−phosphate; G6P, glucose 6−phosphate; F−1,6−BP, fructose1,6−bisphosphatase; F6P, fructose 6−phosphate; PFK, phosphofructokinase; FBA, fructose−bisphosphate aldolase; G3P, glyceraldehyde 3−phosphate; 3−PGA, 3−phosphate glycerate; PEP, phosphoenolpyruvate; PK, pyruvate kinase; PDC, pyruvate dehydrogenase complex; OAA, oxaloacetate; 2OG−DH, 2−OG dehydrogenase; ACO, aconitase; CS, citrate synthase; FUM, fumarase; IDH, isocitrate dehydrogenase; MDH, malate dehydrogenase; SDH, succinate dehydrogenase; TCA, tricarboxylic acid; 3−PHP, 3−phosphohydroxypyruvate; 3−PS, 3−phosphoserine; PGDH, 3−phosphoglycerate dehydrogenase; PSAT, 3−phosphoserine aminotransferase; *AtPSP1*, the gene encoding the Arabidopsis 3−phosphoserine phosphatase (PSP); APK, APS kinase; ATPS, ATP sulfurylase; SIR, sulfite reductase; SULTR, sulfate transporter; SUOX, sulfite oxidase; APR, adenylyl−sulfate reductase; SERAT, serine acetyltransferase; OAS, O−acetyl Ser; Cys, Cysteine.

## Data Availability

Not applicable.

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
