# Peer review of "Overexpression of the Phosphoserine Phosphatase-Encoding Gene (AtPSP1) Promotes Starch Accumulation in Lemna turionifera 5511 under Sulfur Deficiency"

_plants, 2023, doi:10.3390/plants12051012_

Round 1

Reviewer 1 Report

The ResultsDiscussion and Methods sections of this manuscript are fine. The basic point about heterologous over expression of phosphoserine phosphatase enhancing total production of starch in Lemna turionifera under conditions of sulfur depravation is shown and well defended. The description flows logically. Some terminology in the Abstract needs modification.

Line 11.  Change to "Duckweeds are well known for their..."

Line 12.  Give full terminology for PPSB and put the acronym in parentheses.

Line 14.  Change "prompt" to "stimulate".

Lines 15-16.  Change "better detected" to "higher"

Lines 17-18.  Change to "...expression of several genes in starch synthesis, TCA and sulfur absorption, transportation and assimilation was significantly up- or downregulated."

Line 18.  Change "suggested" to "suggests"

The Introduction has several inaccuracies in the first paragraph that need fixing. I attach a suggestion below for the changes to the first paragraph based on the following points:

1. "Duckweed" is the common name for the Family Lemnaceae which has 36 species (the proper reference for 36 is included). When you speak of the Family, the term "duckweed" needs to be given in the plural as "duckweeds". 

2. The plants are not "degraded", they have a "reduced anatomy".

3. "much higher" is ambiguous. "higher" is unambiguous.

4. The sentence in lines 29-30 ("The variety of..... biomass [3]") is factually incorrect as written and should be deleted. It is the composition of the growth medium that is the main factor. 

5. "easily converted" is ambiguous, "converted" is unambiguous.

6. Reference [15] is a non-peer reviewed preprint from 2021 purportedly in the bioRxiv server. However, the DOI provided does not lead to any article. I suggest deleting [15] since reference [17] seems to cover the points the authors want to make for [15] and is a peer reviewed paper by substantially the same authors as in [15]. 

Suggested changes to the first paragraph

"Duckweeds are the smallest flowering plants known to date and have severely reduced anatomies [1]. They are widely distributed all over the world, except in the Antarctic and Arctic regions [2]. There are five genera of duckweeds, Lemna, Spiodela, Landoltia, Wolffia and Wolfiella, with a total of 36 species [Bog, M.; Appenroth, K.J.; Sree, K.S. Duckweed (Lemnaceae): Its Molecular Taxonomy. Front. Sustain. Food Syst. 2019, 3, 117]. The biomass accumulation in some duckweeds is higher than in corn in dry weight (DW)/ha/year due to its asexual propagation and rapid growth [2]. The growth rate is attuned to the richness of the growth medium. It has been reported that duckweed can be converted into four different forms of energy, including bio-oil, natural gas, bio-ethanol and high-value-added industrial precursors, through different conversion technologies [4]. This makes duckweed a promising source of starch and a potential feedstock for production of bioethanol and other biofuels [5]. The starch is mainly synthesized in the fronds of duckweeds, and 3-–75% of dry weight as starch can be accumulated when duckweed is treated by growth regulators, heavy metals, nutrient deficiency or salt stress [6-16]. However, these treatments always inhibit the growth of duckweed. In contrast, sulfur deficiency was found to improve starch yields in duckweed without affecting its rate of growth and biomass accumulation. As a result, sulfur deprivation resulted in higher total starch yield than under nitrogen or phosphorus limitation conditions. The research suggested that the cultivation of sulfur limitation is a potential strategy to prompt starch accumulation in duckweed [17]."

Some additional points:

Lines 80-81.  Change to "....was investigated in duckweed overexpressing AtPSP1....

Line 289.  substitute the full term for "RGR", especially as this is the only time RGR appears in the manuscript.

Reviewer 2 Report

The manuscript “Over-Expression of Phosphoserine Phosphatase-Encoding Gene (AtPSP1) Promotes Starch Accumulation in Lemna turionifera 5511 under sulfur-deficiency” is the second publication of this group of authors on overexpression of a gene involved in serine pathway in duckweed. The study based on a original protocol of agrobacterium-mediated transformation of Lemna turionifera,   further emphases a link between serine biosynthesis and accumulation of starch in a potentially important plant for production of biofuel.

Although the manuscript is relatively well arranged and illustrated, the poor English is the major problem, which should be seriously addressed before the manuscript can be considered for publication.  There are some other flaws that should be corrected:

1.    Line 26-27. “There are 5 genera of duckweed: Lemna, Spiodela, Landoltia, Wolffia and Wolfiella, a total of 38 species [3]”: it is really an outdated choice of a reference.  Currently, only 36 species of duckweeds are recognized; authors can refer to Acosta et al., (2022, Plant Cell) and/or Bog et al (Bog, M.; Appenroth, K.J.; Sree, K.S. Key to the determination of taxa of Lemnaceae: An update. Nord. J. Bot. 2020, 38, e02658)

2.    Figure 1c.  What 18S stays for ? Looks like it relates to 18S ribosomal RNA, should be stated correspondingly.

3.    A big part of the study deals with expression analysis of L. turionifera  genes involved in sulfur assimilation and starch metabolism, analysed by qRT-PCR using gene specific pairs of primers. The authors write that “specific primers were designed from the duckweed genome sequence obtained by RNA- Seq” (line 459-460), providing no further references to the sorce of the gene sequence data, just listing the sequences of primers. The sequences of all mentioned genes should be made available.  

Round 2

Reviewer 2 Report

The expressed concerns have been addressed. 

Author Response

Thank you for your suggestion, the manuscript have been edited by MDPI English office. The new manuscript please see the attachment.
